# Family–SEN School Collaboration and Its Importance in Guiding Educational and Health-Related Policies and Practices in the Hungarian Minority Community in Romania

**DOI:** 10.3390/ijerph20032054

**Published:** 2023-01-22

**Authors:** Beáta Andrea Dan, Karolina Eszter Kovács, Katinka Bacskai, Tímea Ceglédi, Gabriella Pusztai

**Affiliations:** 1MTA-DE-Parent-Teacher Cooperation Research Group, Bonitas Special Education Center, 410032 Oradea, Romania; 2MTA-DE-Parent-Teacher Cooperation Research Group, Institute of Psychology, University of Debrecen, H-4032 Debrecen, Hungary; 3MTA-DE-Parent-Teacher Cooperation Research Group, Institute of Educational Studies and Cultural Management, University of Debrecen, H-4032 Debrecen, Hungary

**Keywords:** special needs teachers, SEN, parental involvement, Hungarian minority, Romania

## Abstract

Special education practice allows for the educational principles of parental involvement, pointing to a common dialogue on health issues and general well-being. Special education professionals primarily empower the families of children with atypical development by relying on the organizational factors of individual support and services. The decision-making/partnering factors of the educational and general health processes, on the other hand, receive less attention. The present study aims to explore the place of the parent–school relationship within the framework of a special educational institution in Romania. Involving Hungarian special education teachers (N = 12) from Romania, we analyze the school involvement of the parents of students with special educational needs in Bihor County, Romania, concerning their children’s academic achievement and well-being. The qualitative research data were recorded through semistructured interviews and were organized through deductive categorization, as well as being analyzed thematically using Atlas.ti. The results highlighted the essential elements of the parent–school relationship, e.g., communication practices, active inclusion programs, support services, and peer acceptance plans. We believe it is important to emphasize that, in the study, families frequently better understood their child’s situation and health-related issues and advocated more effectively for the recognition of their rights. However, as an active player in everyday education, the teacher can better organize development activities for the child’s specific needs and plan complex habilitation/rehabilitation. To sum up, a parent–educator team approach can result in more successful educational and health outcomes, as well as a more accepting social image in the cognitive, emotional, and social development fields.

## 1. Introduction

Individual, social, and societal influences, as well as how they interact with one another, are elements that affect mental health and general well-being. As a result, mental health needs to be understood from biological, psychological, and societal perspectives, and there is a need to simultaneously target numerous complex elements to enhance mental health [1,2]. As in other peripheral European countries, [3,4] in Romania, disability perceptions and equal opportunities issues have undergone significant changes in the last decades in terms of education and the structure of the different care systems. However, a negative evaluation still surrounds the segregated specialized institutions, which are institutionalized and, following the European countries’ model, perform various roles, such as special education, habilitation–rehabilitation, skills development, or even talent management. Thirty-three years have not been able to overcome the traumas of the former black pedagogy [5,6], which have added to the fearsome, even horrifying repertoire of auxiliary institutions. The years following the change of regime from 1999 to 2007, have seen substantial changes [7,8], and the professional orientation of the auxiliary schools moved more and more towards pedagogical and therapeutic activities. Although children with various disabilities are still educated in special education centers, the institutions consider serving and assisting children with special needs as their primary pedagogical goal. In the years following the change of regime, the Hungarian minority with disabilities in Romania continued to be enrolled in Romanian special schools (with the Romanian language) (According to the latest 2022 census data, more than one million Hungarians live in Romania (https://www.nepszamlalas.ro/, (accessed on 10 January 2023)) as involuntary indigenous ethnic minorities [9,10]. Due to the new national borders designated in the peace treaties after World War I, Hungarians in Romania were not placed in a minority situation by their own decision. For several decades, their opportunity to participate in education in the Hungarian mother tongue was limited [9,10,11,12]. This has changed in the last two–three decades, with increased opportunities for learning in Hungarian at all levels of education [9], but this trend was only followed later by special education. It is important to mention that Hungarian-language higher education courses in psychology and special education were only reintroduced into higher education after the 1989 regime change and are only available in the major university centers (counties of Cluj and Mures)). Romania’s school system is still very centralized. Both schools and educators primarily passively fulfill requirements in an overregulated and highly centralized system. After several decades, the local Hungarian lobby succeeded in securing the establishment of Hungarian-language education as an independent institution. Hungarian-language special education centers now operate in three counties: Bihor, Cluj, and Satu Mare. The narratives of the special needs teachers interviewed in this study provide a mirror and an authentic view of what it means to educate a disabled child in Romania.

The international literature has developed the problems that families with children with disabilities have into an intensively researched topic [13,14,15,16,17,18,19,20,21,22]; the results of significant research in Hungarian have been discussed, both from a Hungarian and a Romanian perspective [7,22,23,24,25]. The relationship between families and the cultures of educational organizations has been the subject of rich literature with an international perspective [17,18,26,27,28,29], but it is under-researched in the domestic literature. In Romania, the research on children with disabilities raised in institutions is mainly concerned with the horrors experienced, while the professional discourses of special education are completely absent from these follow-up studies [5,6,30]. Without emergency rhetoric [31], very few people are able to discuss the fate of people with disabilities in Romania, even though there is a positive trend of change in all areas, and confidence-building efforts are beginning to take shape in education. Special education centers and institutions providing care for children with disabilities are, through their additional services, fulfilling complex tasks that contribute to education reform efforts. The partnership with civil and church-run organizations, especially for the Hungarian minority in Romania, is a milestone, turning inclusive education into a successful practice.

A systematic approach to the topic is not without precedent in the literature. Najev Čačija et al. [3,4]–based on the work of Kyriazopoulou and Weber–distinguish three system levels: the macro (legal and national context), mezzo (practice in school), and micro (classrooms and the interaction between students and teachers). Parents and families play an important role at all three levels [3,4].

Environmental factors feature prominently in the study of the developmental stages of the individual. In the present study, Uri Bronfenbrenner’s ecological model was chosen as the theoretical framework, which analyzes the environmental systems that influence the development of individuals and the systems belonging to it [32] based on the philosophy that individuals are strongly connected to the environment they live in, which impacts them. Bronfenbrenner argues that individual behavior can be understood by looking at four environmental factors in a system. Layering, in the form of onion skins [33], is interwoven around the individual: (1) the microsystem (the immediate environment), (2) mesosystem (the expanding places and groups of the wider social structures), (3) exosystem and the institutional systems, as well as the value and norm systems, and (4) the macro system. These different systems can be considered nested system layers. The developmental trajectory of the individual is closely linked to the structures of the environmental factors that accompany the individual from the moment of birth. Concerning persons with disabilities, the ecological model also advocates a shift away from a medical/pathological perspective, as it views the person as a complex process and provides a systematic view of psychological development. In looking at special education processes, the ecological model views the child with special educational needs in the light of context [32,34,35].

The microsystem includes the interaction processes between the child and the people, objects, and places in the immediate environment, and school and home belong to this level [34,36], as do the activities and interpersonal relationships taking place in this immediate environment. Direct interactions can be detected here, composed of levels of persons, objects, and symbols. Bronfenbrenner evaluates this level as a dyad since it is a relationship between two individuals most of the time: mother and child or parent and child. A third person may enter this system or any other individual belonging to the child’s immediate environment, who may bring destructive or constructive qualities [37]. 

The mesosystem is a structure built on the microsystem, which may include school–home relationships, with circular causality, and communication practices and information exchange are also influenced by the interdependent structures. 

The stratification of the exosystem includes a myriad of social organizations, such as the media, government organizations, parents’ workplaces, and a range of other service providers, which indirectly but significantly influence the developmental trajectories of children [36,38]. The macro system is a set of ideologies, cultures, and subcultures continuously shaping and molding the child’s worldview [37]. 

The author later extended the four layers outlined above with a chronological system (5) [39]. This is the last step in ecological systems theory, which makes the individual dependent on all environmental factors, including life cycle shifts and the arc of historical drift. In Romania, the most significant change in the chronosystem was the 1989 revolution, which also pushed disability modeling in a new direction.

Although ecological systems theory emphasizes the influence of environmental factors, the Russian psychologist does not ignore the influence of biological factors when studying individual development. Bronfenbrenner and Ceci [39] developed the bioecological model of development, acknowledging the combined influence of genetics and environmental factors, arguing that harmonious individual development can only be optimally balanced concerning genetic and environmental influences. Ecological systems theory has not only initiated a paradigm shift in psychology but has also led to major paradigm shifts in the scientific approach to psychopathology and psychopedagogy [40,41]. Ecological systems theory emphasizes both intra- and interpersonal influences, making it the most influential theoretical framework that has also greatly reshaped modern disability theory. It positively impacts how intervention programs and habilitation/rehabilitation development plans are developed, leading to the development of educational and organizational factors in the educational process of children with special educational needs towards inclusive placement [42]. 

From the ecological systems theory perspective, individuals and environmental structures at different scales are unified systemic entities that interact with each other. Systems can be organized hierarchically or as interconnected and evolve subsystems and higher-order systems [43,44]. All system theories provide a framework for the disciplines, striving for unity and, more specifically, examining the whole in terms of its processes of connection, interaction, self-organization, and cognition. Open systems constantly interact, regulating their state and controlling their existence. Elementary components of open systems include input, change/effect, output, and feedback [43]. Changes/effects in each structure affect the whole system [44].

The feedback loop is a graphical representation of the input and output of systems; in this case, the relationship between the educator and the child with a disability concerning the school system. The child’s behavior is shown as an output factor, and the educator’s response becomes the input factor. The system’s steady state depends on changes in the negative and positive feedback loops. Negative feedback can make the system unstable, while positive feedback can strengthen it [45]. Ecological systems theory also emphasizes the role of assessment and monitoring protocols in educating pupils with special educational needs, building on the development of individual skills and abilities and the optimal presence of environmental factors.

The relevance of the close and wide environment is crucial in the case of children with special needs. They often experience more barriers and developmental delays, which require serious attention and treatment. Therefore, their support is critical in their development. We usually focus on the close environment, including the child’s family and the special education teachers and educators working with them. Additionally, the connection between the family and the teachers (and the organizations e.g., nursery or school) must be emphasized. In this paper, we aim to focus on the manifestation and characteristics of this connection system.

## 2. Materials and Methods

### 2.1. Research Questions

The research problem of the present study is the place and role of the parent–school relationship in the field of Romanian special education, and two research questions are raised around this problem:What challenges do special educational needs teachers face in terms of the practical aspects of family involvement?What are the most urgent challenges to strengthen the family–school relationship?

### 2.2. Methods

The data were analyzed using constructivist grounded theory, which is an inductive, iterative method. With our research, we add a new element to the trend characterized by Najev Čačija and their colleagues [3,4], in which they conducted research in three peripheral European countries using the GT methodology with the aim of generating a theoretical model based on the experiences of experts. This time, we are building on the experiences of special education teachers. To the best of our knowledge, no such research has yet been carried out in relation to Hungarian-language special education in Romania. Using grounded theory (GT) [46,47], we try to move away from our knowledge of the subject (one of the authors of this research is a practicing special education teacher; the authors consider it important to explore the subject, yet they consider it important to stay objective and explore it with a researcher’s eye), in order to formulate our theoretical justifications, starting from the data obtained from the interview analyses, and moving steadily along the gradients of abstraction. The semistructured interview questions serve only as a starting point, providing space for free-associative reflection to discuss issues important to the interviewees and relevant to the topic [48,49]. At the same time, a deductive category analysis is used to build up the analysis. A category-driven textual interpretation based on qualitative grounds will be emphasized, also highlighting the possibilities for feedback and intersubjective testing using the ATLAS.ti software (22.2.5 Student version). Drawing on the methodology of GT, a constructivist meaning-making process will be followed, thereby focusing on the “what” and “how” questions to explore the patterns of engagement of special needs teachers and families with children with special educational needs and disabilities in the school. We used GT techniques to identify themes and patterns, as well as to create categories of reasons for the family-SEN school collaboration and its importance in guiding educational and health-related policies and practices in the Hungarian minority community in Romania that the participants suggested. The interview data were later analyzed by two different researchers who approached the material using the GT method. In the data, incidents are identified and coded. Then, the initial codes were compared to other codes. The codes were then grouped into categories. The researchers compared incidents in the same category to incidents in different categories. Future codes and categories were compared to one another. The new data was then compared to the information gathered earlier in the analysis phases. This iterative process involved both inductive and deductive reasoning. Inductive, deductive, and abductive reasoning were all used in the interview analyses.

### 2.3. Participants 

Theoretical sampling was applied during data collection [50,51]. In the beginning, we defined a wider range of possible interview subjects based on expert criteria, and then, simultaneously, with the analysis of the interviews, additional subjects were selected from this range on the basis of emerging concepts. Sampling was guided by the clarification of the categories revealed during open, axial, and selective coding.

The expert criteria defining the wider range of possible interviewees were as follows: (1) the subjects must be employees of the Hungarian-language Special Education Center in Bihar County, which operates in 3 cities and welcomes students with disabilities and SEN students aged 4–18; (2) they should be certified special education teachers; (3) a minimum of 5 years of experience in the field; (4) work with the head of the class, so they know the students’ families better; (5) they should have a higher education diploma in Hungarian so that the professional terms that arise during the interview can be understood in the same way as the interviewer, and the analysis can be carried out on a homogenous linguistic basis.

During the analysis, when we reached theoretical saturation regarding the fundamental question of the research, and the emerging theory was properly grounded in the data [50,51,52,53], we wrote the present study using 12 interviews. Based on the literature, saturation can be achieved with 9–10 interviews [52,53]. Our analysis is also characterized by the factors that can reduce the size of the sample required to achieve saturation. These are the following: the subjects form a homogeneous group and are truly experts on the subject; the researchers know the relevant literature well and have sufficient knowledge of the topic; the topic is less sensitive for the subjects (e.g., nonpersonal issues); the research question is sufficiently narrow; the extracted data are sufficiently rich, content, and informative [52,53]. Since the theory generated during the analysis can be further enriched by refining additional categories, we will continue data collection in the future.

All participants volunteered to take part in the interviews. Although a professional selection process preceded the selection of the teachers to be involved in the semistructured interviews, the researchers considered it important that no pressure or top management constraints should be a factor. All participating teachers were psycho educators (the professional designation used in the Romanian education system)/special education teachers with several years of experience in special education. Three teachers have 3–5 years of experience, five teachers have 10–15 years of experience, and four teachers have 20–25 years of experience in special needs education. Twelve special education teachers were interviewed in 45 min and 1.5 h time blocks during the 2021 and 2022 school years. All the participants are female. Concerning the age range, four teachers were aged 24–29, six were aged 30–39, and three were aged 40–55. The interviews were recorded as audio files, which were converted into text with the help of the third author and imported into ATLAS.ti 22.2.5 Student version. The semistructured interview schedule was structured around 11 topics. The interview outline included several main questions per topic and the supporting and clarifying questions. When recording the interviews, the interviewer asked all the main questions in the order given. The interview schedule was not specifically designed for special education teachers but was recorded as part of a larger study of teachers with this particular target group. The research group aims to increase the parental engagement of teachers, and the interviews we analyzed were conducted in the initial exploratory phase of this research. 

The interview transcripts provide a detailed description of the family–school-related involvement habits of the special needs teachers involved in the research, and their personal experiences, thoughts, and feelings on the topic. All interviews were recorded in face-to-face interviews, and the subjects’ informed consent was also recorded in writing. During the processing of the interviews, only the anonymized transcript was analyzed, and the proper names (personal names or place names) were omitted.

## 3. Results

The transcribed interviews were analyzed through an open axial coding process, thereby organizing the data and structuring them around literature-based terms following the grounded theory methodology [54]. The resulting data were consolidated and interpreted following a deductive coding process, grouping them into two main themes and 16 subtopics.

The next step was to merge the code families and organize them into thematic units along the lines of ongoing collation and analysis [48]. Once the coding process was complete, two authors collated the codes and integrated them into a thematic unit within a broader interpretive framework. The broader axial coding process is described by Glaser and Straus [47]. The second author then reviewed the category systems developed, and a consensus was reached on the breakdown of the themes and subtopics. This resulted in merging 69 codes into 32 main codes, which were grouped into two main topics and 16 subtopics. 

The grounded theory also establishes the importance of highlighting contextual influences, seeing the personal characteristics, family relationships, work environments and community influences, cultural values, and national characteristics (in this case, the impact of the Romanian school system on the education policy of the Hungarian minority), which also helped to reveal the sociopsychological influences [51]. Taking all this into account, we have identified and thematically ordered the characteristics of the micro and macro environmental impacts in the interview texts (see Table 1).

The thematic units that emerged from the interview analyses, following the pattern of ecological systems theory in the international literature, also show the converging findings of several curricular studies on the topic of the family–school–community triad [55,56,57,58]. In the following, we present the process of the data-driven analysis and the construction of the categorization items by quoting the interviewees. Two main categories emerged from the data. These were grouped around the challenges and tasks to solve in the parent–school relationship, and subcategories were assigned to the main categories (see Table 2).

### 3.1. Main Category—Challenges

In the analyses, we investigated the barriers that special needs teachers perceive to be related to parental involvement. Special education teachers talked about the nature of the problems encountered and their causal factors, with a focus on the different parenting principles, lack of time, and inflexible and uncaring parenting:


*“There should be an openness on the part of the parents, a greater openness which they don’t have… they don’t have. So it’s a very fast-paced world, and maybe that’s why they don’t have the time to have a more open, closer relationship with the school.” 1:21.*



*“I see that there is no system in children’s education, no rules at home… I try to set up rules at school, but it’s very difficult to keep them because, at home, the children are used to the fact that there are no rules and no consequences, they always do what they want, and because of that, it’s very difficult for the children to adapt to school… and on the other hand, children like home better because they know that there… there are no consequences for what they do; they can do anything, anytime, because there is no system, no rules.” 4:14.*



*“I see in many cases that they really let the child do what they want, and that makes it harder for me, and it affects about 50% of my class, that I try to use a parenting technique, but it’s not useful if the child does what he/she wants at home. He/she wants to get it at school, too, even with aggression, tantrums or physical abuse. So, unfortunately, many parents let them do it because it’s easier that way. Or they take the child home, put them in front of the TV, and they’re fine. Many parents, when they have a child like that, get tired of parenting and try to put the deflect on the institution and the teachers, and they don’t understand that they should be a partner in this, and then I alone won’t be able to do miracles.” 5:5.*


In many cases, the negative tone of the speech is also one of accusation mixed with incomprehension:


*“Rather, I’ve noticed they’re happy if we don’t have to talk about anything because then they think there’s no problem, and that’s okay.” 4:19.*



*“For example, where the parents are illiterate, there is no system.” 7:4.*



*“Now, if there is no contact, the parent doesn’t know about it, no one is held accountable for anything, and everyone does what they want.” 7:33.*


However, in almost all cases, the interviewees conclude the negative thought they have started with understanding and empathy:


*“There are things like that, which are… no no, we can’t communicate them without offence, sometimes you have to listen… and then…” 5:24.*



*“They are not angry with me; specifically, they are angry with the government and the system, it was just me who was there as the mediator.” 6:14.*


The family–school relationship of special education teachers can best be understood from the ecological systems theory approach since interview analysis can slip in the wrong direction without understanding the ecological models and their effects. We need to be able to include several social variables in the study of disability perceptions among the Hungarian minority in Romania. The possible Romanian ecological system-theoretical factors include deprivation and extreme poverty, socioeconomic status, culture, stigma, identity, and the linguistic socialization milieu. The interviewees also emphasized the importance of families’ economic and symbolic capital, stressing the overburdening of mothers, who are the primary caregivers and who maintain contact with schools. In Romania, gender roles are evolving in line with the European trend, with fathers becoming involved in the upbringing of their children. However, mothers are responsible for the care and daily tasks of upbringing:


*“It’s absolutely the mothers who are more likely to call me, who are more likely to write a message or reflect in the group. There are some fathers who walk with the child and bring it to school, but not particularly. They don’t open up to me either, and like, …, they think that it’s the woman’s job, I think. I think {in this city}, this is the mentality that the mother raises the child. So it’s still here, it’s still strongly typical, this old mentality.” 5:15.*



*“Rather the mothers. So it’s usually the parent who keeps in touch with the school, who goes to the parent-teacher meetings, who asks how the child is doing, and it’s usually the mother.” 7:14.*


Aggressive behaviors also have a significant impact on the development of the teacher–parent relationship, as the lack of human resources often places the burden of dealing with behavioral problems (for various reasons) on the special education teacher:


*“My most challenging student, well, he was an autistic student, who unfortunately was very aggressive, he would attack me, he would attack his peers. And yes, his mum, anything I said to her, she perceived it as an attack, and it was very…, well she found it very difficult to communicate between us. It was always the given teacher who was the bad one, because we can honestly say that it’s very difficult to work with the student… Communication was very difficult with him; he often took what I said as an attack, but I never really attacked him, but he had problems with the situation, with life, and again, things were taken out on me, just like in previous years with the teacher who was teaching him. Well, after a while, you get used to it and get used to it.” 5:28.*


The defining characteristics of the parent–school relationship can be enhanced or undermined by social acceptance. In Romania, special education centers are segregated institutions, and inclusion in mainstream educational, cultural, and recreational programs is a challenge that emerges in the interview transcripts:


*“There’s no… there’s no specific budget to… to organise programmes for parents.” 7:6.*



*“Because they are children with special needs, they cannot be taken everywhere or are not always welcome…” 7:7.*



*“You can take them to the puppet theatre, uh… to church; so we have… we have partners… who are open to our children, but… for example, a… a theatre performance, an autistic child might not be… welcome, because they’ll … they’ll shout, they’ll stand up if they don’t like it…” 7:8.*



*“It is the only Hungarian-language special education centre in the county, so… I think the school has a significant role in ensuring that children with disabilities are… properly… educated. And… I think there are… quite a lot of steps… being taken today to make sure that they know about us and that… GPs or even… the school principals in the county… know that if they have a student like that, they should refer them to us…” 7:9.*


### 3.2. Main Category–Tasks

The outlines of the tasks based on effective school–family partnerships were considered an important part of the interview analysis, which the special needs teachers mention mainly in terms of organized and thematic dialogues and their potential positive effects:


*“The one who needs the most attention in our house is a little boy with epilepsy. However, in his special case, his mother works and thank God, we can talk to him, so he is very open, he even asks for advice, and we also discuss what happened at home, how he behaved, and what medication he is on. So with her, I would say we have a very good relationship.” 6:23.*



*“There are cases like that, yes, when the parent doesn’t hear what you’re saying but is very offended by that. You have to accept that, and then move on and, er, go on in the same way, be nice in the same way, smile at them in the same way, and then the situation is resolved. Because they are, I have noticed that most of the time when a parent attacks, it’s not you they have a problem with; it’s themselves, it’s life, it’s the situation, and you’re the only one there to help. So… But the cases are, uh… In most cases, I find that parents are quite receptive, quite helpful and try to be there for me and help me, even if they’re in a situation where they’re pushing me to do things like testing the kids.”*


At the same time, there is an emphasis on educational blueprints for acceptance, which teachers frame in terms of benevolence and helpfulness, highlighting the positive impact of ongoing communication and the effects of trust and commitment:


*“Mutual respect characterises this relationship. So I accept how they are and how they relate to the child, and then they accept that I try to do my best even if sometimes mistakes slip in because mistakes have slipped in unintentionally, and we have treated each other as human beings. I think that’s the most important thing.” 6:14.*



*“As a parent, it means that my child is different from other parents with children. And then the parents told me how difficult it is to go out on the street and then go into a shop, but there’s no transport, but if you’re in a wheelchair, for example, there’s no security, if he can’t look away or the shop assistants won’t tolerate the child touching this, touching that, or shouting. He makes a circus, and then the whole store gathers around and watches, it’s hard enough. Now we also have a disabled child in our family, and I think that from that point of view, we should have something that the parents of these disabled children can meet. We should discuss how each of us takes these obstacles, and it should be a topic that… what do you do when ten children do it, and yours is the eleventh who doesn’t know it. How do you explain this to them, or how do you go through it? I think there should be something like that and where they can go for help, for example, if they have a problem or whom they can trust the child with if the mother wants to get out, for example.” 6:21.*



*“I think after a while, when you’ve been with a class a lot, you get a sense of how much communication and support the parent needs. In fact, a bit of counselling is part of our profession. Because many parents are really clueless, and it’s good to have support. For example, I’ve always considered myself to be such an empathetic person, and so… well, I can empathise with their situation.” 5:8.*


Parents of children with disabilities face many significant difficulties in educating their children, as it’s an everyday task to cope with the constant care, provision, education, and health issues, which require much more time and energy, and special education teachers recognize this and see their role as one of encouragement, social–emotional support, and guidance.

## 4. Discussion

The present study examines the relationship factors between special education teachers and parents in the Hungarian minority in Romania. According to our findings, this relationship was strongest when parental involvement was defined as positive for family–SEN school collaboration. The expectations of special education teachers reflect educators’ beliefs and attitudes toward school and SEN education in Romania. Several educational and health-related topics were examined in the interview analyses, including those factors related to current challenges and tasks. The research involved a relatively small number of interviews (N = 12). It focused only on one special education center in Bihor County, but we interviewed special education teachers from three cities (one county seat and two smaller block Hungarian minority cities).

In this study, we draw conclusions based on the first research question, “What challenges do special educational teachers face in terms of practical aspects of family involvement”. When examining parental involvement, the participating teachers encountered external or internal barriers (organizational, structural, or interpersonal) and conflicts for a variety of reasons. The second research question, “What are the most pressing challenges to strengthening the family–school relationship?” We reached a number of different conclusions. Several social variables must be considered in the study of disability perceptions among Romania’s Hungarian minority. Deprivation and extreme poverty, socioeconomic status, culture, stigma, identity, and the linguistic socialization milieu are all possible Romanian ecological system-theoretical factors.

As Amor et al. [59] state, although several pieces of previous research have focused on the development of theory and describing practices and attitudes, which are inevitable steps in advancing inclusive education, further investigations are of paramount importance since good international practices are still lagging behind in both the Romanian and Hungarian context. The parent–teacher relationship is an important element of education and general health policy planning in Romania, but the special needs education dimension is not given much attention. This is why the results of the interview analysis are relevant, as they provide a more detailed insight into the education and development process. The special education teachers interviewed are a credible reflection of the current situation in Romania, both in terms of the problems and challenges they face and the challenges they are trying to address. The most pressing issue is the creation of resources and the need to address the problems of families’ disadvantaged situations. There are several challenges, but teachers could be more effective in representing the interests of parents and pupils in a more accepting society through acceptance and support. However, we must emphasize that this is a global issue that is not limited to Romania [60,61,62]. For a more inclusive, tranquil, equitable, and prosperous society, education is crucial to achieving the sustainable development goals (SGDs) [63].

Special education teachers have a key role in parent–school relationships, as best illustrated by Bronfenbrenner’s ecological model, as they are responsible for the design and implementation of educational, health, and developmental programs at the micro level but also play an important role in the day-to-day interactions between families and children. The latter can be facilitated by other helping professionals working in and out of school [64]. At the micro level, they are also responsible for implementing education policy plans. Parental involvement must be defined and evaluated while considering the cultural and individual characteristics of parents raising children with special educational needs. Additionally, the peculiarities of inclusive education and the characteristics of international, collaborative research should be emphasized when following the conclusions of Amor et al. [59]. These factors are crucial for a long-term perspective as well, e.g., focusing on the educational pathway of children with special needs [65].

## 5. Conclusions

The study involves teachers in different, specific situations regarding Hungarian teachers in Romanian SEN schools. Both a theoretical and a practical conclusion can be drawn from our research. The ecological model served as the theoretical foundation for our research, and it proved to be consistent. The SEN child is at the center of the model, and the parents and teachers who operate in this multiplex environment make up the microsystem. The meso-, exo-, and macrosystems, which include the local community, media, and culture, have an impact on the activities and interactions of the members of the microsystem, as do other levels outside of it. The SEN child is, thus, at the center of all levels, and his or her development, reaching the upper limit of his or her ability to be educated, should be the common goal of all actors in the system. Starting from this theoretical conclusion, we can develop the intervention steps at the policy and practice levels that we outlined in the practical field [3,4].

Toward sustainable development goals, policies should work with communities, parents, teachers, and other educational personnel to ensure quality educational goals, and it should be supplemented by measures to assist families in coping with isolation and stress, such as identifying and promoting practices in behavioral change communication around the SEN school system, social inclusion, and family engagement. The focus in special education is on learning and growth, but adherence is also important in development, as it is the responsibility of the parents. Teacher education and the professional development of teachers are particularly important to the success of the implementation of inclusion [3,4]. They should prepare teachers to communicate with parents on behalf of the child with the appropriate empathy and knowledge. According to our research, this is also difficult because most SEN students come from low-income families, and in many cases, the parents themselves have SEN, making communication and co-operation even more difficult.

Social acceptance and inclusion are also influenced by actors within the meso, exo, and macrosystems. The rights of the Hungarian-speaking minority must be guaranteed to receive a good quality education in their mother tongue. On the one hand, societal attitudes toward exclusion or inclusion affect families’ daily lives and parents’ attitudes toward their children. Additionally, the rights of the Hungarian-speaking minority must be protected from discrimination. Moreover, this is a question of regional and international policy. Our research has led us to the conclusion that, without it, special education teachers may encounter difficulties at work that they would not encounter if these rights were ensured. Excessive centralization makes it difficult to respond to issues affecting local communities in a responsive and timely manner at the mesosystem level. A more collaborative organizational structure produces better results than a traditional bureaucratic and inflexible school environment for parent involvement programs [3,4].

In order to achieve the SDGs, Romania must broaden its inclusive education perspective and practice. The current study could be expanded by conducting additional interviews, introducing participants to broader concepts of collaboration, and asking participants to generate additional ideas about family and SEN school collaboration. We also believe that future research should contribute to a more positive relationship between SEN schools and families. Efforts should be made to bridge the gap between SEN institutions and families, giving parents and teachers the opportunity to build a positive relationship while addressing SDGs.

Although the local requirements for family–SEN school collaboration found in this study are context-dependent and come in a variety of combinations, they share some characteristics with those found in GDSs in terms of both their criteria and components. The educational system, inclusive policies, accessibility to educational, habilitation, and rehabilitation services, general health concerns, and, for the current generation, freedom of choice and action are a few of the local criteria that were identified in our study but did not directly depend on local ecosystem services. Other criteria, however, such as SEN school models that provide inclusive educational opportunities, were specifically linked to local ecological services.

## Figures and Tables

**Table 1 ijerph-20-02054-t001:** Micro and macro environmental impact factors.

Individual	Family	Institution	Community	National
Motivation	Support	School and organizational resources, professional socialization environment	Eligible services, subsidies	Education policy trends
Years of experience/qualification	Structure	Availability of special services	Accepting differences	Social and cultural (multicultural) values
Interpersonal, social skills	Capital (economic and symbolic)	Cooperation strategies for organizational factors	Intergenerational involvement	Educational guidelines, policy decisions promoting inclusion

Source: own edition.

**Table 2 ijerph-20-02054-t002:** Main and subcategories.

Main Categories	Interpretative Framework	Subcategories
Challenges	External or internal barriers (organizational, structural, or interpersonal) and conflicts for various reasons experienced by the participating teachers when examining parental involvement	Different educational and health-related principles
		Lack of time
		Lack of a supportive family environment
		Disadvantaged situation
		Inflexibility
		Unrealistic expectations
		Lack of resources (lack of support systems)
		Burnout (too many sources of stress) -Overworking mothers
		Segregation
Tasks	Subject-specific educational practices and organizational policies that have a positive impact on the relationship patterns between parents and teachers	Cooperation
		Encouragement
		Feedback /positive/negative
		Time
		Social/emotional assistance
		Guidance
		Creating resources

Source: own edition.

## Data Availability

Data are available only on request due to ethical restrictions.

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
