# Peer review of "Family–SEN School Collaboration and Its Importance in Guiding Educational and Health-Related Policies and Practices in the Hungarian Minority Community in Romania"

_ijerph, 2023, doi:10.3390/ijerph20032054_

Round 1

Reviewer 1 Report

This is an interesting paper on the context and the role of special education in the Hungarian-language special education centers in Romania. The authors probably presuppose that everyone must be aware of the fact that special education in the isolated institutions is considered to be an outdated, if not even a cruel method, inflicting on the human rights of students with special needs. There is a whole literature on inclusive education, which should be, at least, presented in very broad terms. A good starting point might be a systematic review, such as: 

Amor, A. M., Hagiwara, M., Shogren, K. A., Thompson, J. R., Verdugo, M. Á., Burke, K. M., & Aguayo, V. (2019). International perspectives and trends in research on inclusive education: A systematic review. International Journal of Inclusive Education23(12), 1277-1295. 

Authors make an interesting attempt to use the grounded theory, within the micro- and macro-levels of special education. Similar topic has been analyzed by following authors, covering the issue of inclusive education at micro-, mezzo- and macro-levels of educational systems in peripheral EU member countries:

Najev Čačija, L., Alfirević, N., & Jurić, S. (2020). Towards an Identification of Critical Success Factors for European Inclusive Education. In Educational Leadership, Improvement and Change (pp. 139-154). Palgrave Pivot, Cham.

Najev Čačija, L., Bilač, S., & Džingalašević, G. (2019). Benchmarking Education Policies and Practices of Inclusive Education: Comparative Empirical Research—The Case of Croatia, Italy and Portugal. In Educational Leadership in Policy (pp. 117-134). Palgrave Macmillan, Cham.

Authors might also wish to provide a more generalizable framework and identify specific future research topics, based on their application of grounded theory.

Author Response

Dear Reviewer,

thank you very much for your valuable review. Based on your suggestions, we carried out the following modifications:

  • references suggested by the reviewer (Amor et al., 2019; Najev Čačija et al., 2019, 2020) were incorporated into the paper.

We hope that the modifications are appropriate.

Kind regards,

The Authors

Reviewer 2 Report

Your study entitled Family- SEN school collaboration and its importance in guiding educational and health-related policies and practices in the 3 Hungarian minority community in Romania. The research topic is related to the goal of this journal. There are some comments on your study as follows:

1.     The ecological systems theory should be added more content to explain and offer a comprehensive literature review on it.

2.     The research method should offer more detail on why and how to conduct this study. The detail is not very clear.

3.     Please add more participants in this study. The number of participants is too small.

4.     The discussion part need more explanation to express your major findings and results.

Author Response

Dear Reviewer,

thank you very much for your valuable review. Based on your suggestions, we carried out the following modifications:

  • we added more content to the ecological systems theory to explain and offer a comprehensive literature review on it
  • the applied research method is now also detailed to provide a more complex perspective
  • we clarified why did we have a relatively low number of participants
  • more explanation is given in the discussion to express your major findings and results.

We hope that the modifications are appropriate.

Kind regards,

The Authors

Reviewer 3 Report

line 60, it is suggested to add the reference to https://pesquisa.bvsalud.org/global-literature-on-novel-coronavirus-2019-ncov/resource/pt/covidwho-1196106 as they add the perspective of social inequalities that are encouraged during confinement due to the absence of a specific plan for the provision and training of users.

In the summary section, it can be understood that it is somewhat ephemeral given the length of the results. It is suggested that, in addition to the commentary on the qualitative aspect of the results (in relation to the initial objectives), the research prospective that the research carried out may yield should be added. Another aspect to be taken into account is the assessment of how the UN Sustainable Development Goals (SDGs) can have an impact on policies to combat social, academic, economic and gender inequalities from the perspective of the authors. 

Author Response

Dear Reviewer,

thank you very much for your valuable review. Based on your suggestions, we carried out the following modifications:

  • we detailed how the UN Sustainable Development Goals (SDGs) can have an impact on policies to combat social, academic, economic and gender inequalities from the perspective of the authors.

We hope that the modifications are appropriate.

Kind regards,

The Authors

Round 2

Reviewer 2 Report

Thank you so much for your revision. Good job!